# Dwarf planet (1) Ceres surface bluing due to high porosity resulting from sublimation

Stefan E. Schröder [1]✉, Olivier Poch [2], Marco Ferrari [3], Simone De Angelis[3], Robin Sultana[2], Sandra M. Potin[2], Pierre Beck[2], Maria Cristina De Sanctis [3] & Bernard Schmitt [2]

The Dawn mission found that the dominant colour variation on the surface of dwarf planet Ceres is a change of the visible spectral slope, where fresh impact craters are surrounded by blue (negative spectral-sloped) ejecta. The origin of this colour variation is still a mystery. Here we investigate a scenario in which an impact mixes the phyllosilicates present on the surface of Ceres with the water ice just below. In our experiment, Ceres analogue material is suspended in liquid water to create intimately mixed ice particles, which are sublimated under conditions approximating those on Ceres. The sublimation residue has a highly porous, foam-like structure made of phyllosilicates that scattered light in similar blue fashion as the Ceres surface. Our experiment provides a mechanism for the blue colour of fresh craters that can naturally emerge from the Ceres environment.

[1] Deutsches Zentrum für Luft- und Raumfahrt (DLR), Berlin 12489, Germany. [2] University Grenoble Alpes, CNRS, Institut de Planétologie et d'Astrophysique de Grenoble (IPAG), Grenoble 38000, France. [3] Istituto di Astrofisica e Planetologia Spaziali-INAF, Rome 00133, Italy. ✉email: stefanus.schroeder@dlr.de

When Dawn arrived at dwarf planet (1) Ceres, it quickly became clear that the dominant colour variation over the surface is a variation of the spectral slope from the visible to the near-infrared (near-IR)[1–4]. Material that we refer to as blue is characterized by a negative spectral slope, both in absolute and relative sense. The average spectrum of Ceres is almost flat and featureless from the visible up to 2.7 µm, where a complex of bands beyond 2.7 µm indicates the presence of ammoniated phyllosilicates[5], likely of the smectite group[6]. The widespread distribution of these aqueous alteration products points to a global distribution of liquid water in the interior of Ceres in the past[7]. Presently, water ice appears to be abundant just below the surface[8–10], and the crust may hold up to 25% of water ice by volume[11]. Blue material on Ceres is uniquely associated with fresh features, predominantly young impact craters[4,12,13]. Some of the strongest blue signatures are associated with Haulani, a 34 km diameter crater located at 10. 8°E, 5. 8°N, which can be clearly recognized at the centre of the global map in Fig. 1 (ref. [4]). Haulani is one of the youngest craters on Ceres, with an estimated age of 2 Ma[12,14], and both its interior and ejecta are blue. Figure 2 shows Haulani as it appeared to Dawn's Visual Infrared Mapping Spectrometer (VIR)[15]. The plot shows the reflectance spectra at two locations, one in the blue interior and the other outside the crater at a location about two crater radii distant, where bluing is not evident[4]. The ratio spectrum reveals that the blue nature of Haulani is characterized by the introduction of a negative spectral slope over the visible and near-IR range, up to 3 µm. Beyond 3 µm the spectrum is increasingly dominated by thermal emission. The blue trend in the ratio spectrum in the thermal IR reflects the fact that Haulani is colder than its surroundings[16].

The origin of the blue colour is not yet understood. On comet 67P/Churyumov-Gerasimenko, a bluing of the spectral slope is interpreted as meaning enriched in water ice[17], but water ice is not stable on the surface of Ceres[18], apart from permanently shadowed regions on the poles[19]. Schröder et al.[4] proposed an explanation on the basis of an experiment by Poch et al.[20], in which sublimation of a mixture of montmorillonite and water ice resulted in an extremely porous sublimation residue that was bluer than the original material. When smectite phyllosilicates like montmorillonite are mixed with liquid water, hydration of the inter-layer space will lead to partial delamination, resulting in suspended nano-sized platelets and aggregates thereof. Freezing the suspension leads to the growth of water ice crystals, with platelets concentrating at the crystal interfaces[21]. Subsequent sublimation replaces the ice crystals with voids, and only the phyllosilicate platelets remain to form a porous network, or foam. Poch et al. tentatively attributed the bluing in their sublimation residue to Rayleigh scattering by sub-µm-sized scattering centres in the foam.

On Ceres, impacts would mix the phyllosilicates on the surface with water ice below to create ejecta of icy mud, which sublimation residue could be similarly porous and blue[4]. Stephan et al.[13] instead argued that it is the impacts themselves that pulverize the phyllosilicates into dust, out of which aggregates spontaneously form to scatter light in Rayleigh fashion. The latter model is based on grinding experiments with antigorite by Bishop et al.[22]. In both explanations, the bluing is attributed to Rayleigh scattering by fine phyllosilicate structures, but they differ in how these structures are formed. Bluing due to the presence of salts, which have been identified on the surface of Ceres[23], is problematic because it would be accompanied by a strong increase in brightness. Rayleigh scattering in a particulate surface was first proposed by Clark et al.[24] to explain the blue reflectance spectrum of the icy surface of Dione. Models incorporating Rayleigh scattering have since been developed to interpret observations of bluing on other planetary surfaces[25,26]. The question remains how structures of ultra-fine phyllosilicate particles come into being. It is the goal of this paper to demonstrate that such structures can form naturally and in large quantities under Ceres conditions.

Here, we show experiments in support of the hypothesis that the blue colour results from a high porosity of the surface, induced by sublimation of an ice–phyllosilicate mixture produced by impacts[4]. We create ice particles with a compositional analogue of Ceres surface material and sublimated the sample under pressure and temperature conditions representative for Ceres. After completion, we determine the physical state and mechanical properties of the sublimation residue. Reflectance spectra of the sample are acquired throughout the experiment.

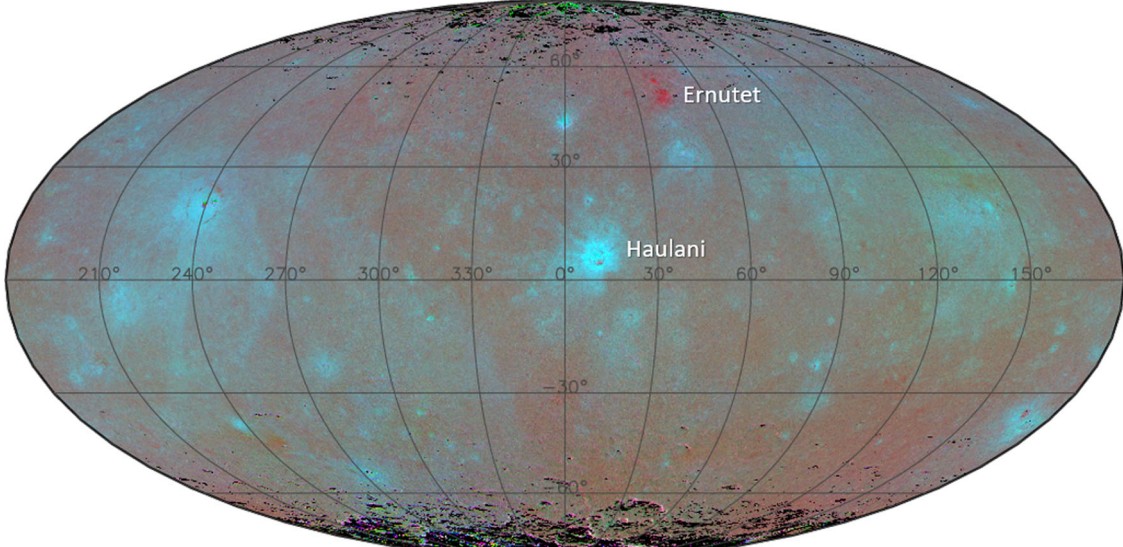

**Fig. 1 Global distribution of blue material on Ceres, with longitude (horizontal) and latitude (vertical) indicated.** Shown is a global false colour composite with the photometrically corrected reflectance ratios of 965/749, 555/749, and 438/749 nm in the RGB colour channels, respectively[4]. This colour coding scheme emphasizes variations in spectral slope, with blue and red areas having a negative and positive spectral slope, respectively. One of the bluest areas on Ceres is Haulani crater, whereas the reddest area is associated with Ernutet crater[53]. The horizontal extent of the map is 3030 km.

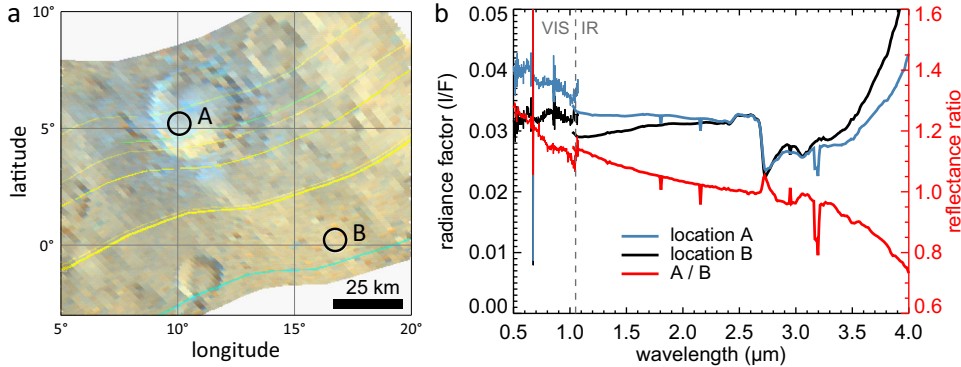

**Fig. 2 Bluing in Haulani crater. a** A map of Haulani and its surroundings from VIR cube 494731110 (red: 2.25 μm, green: 1.75 μm, blue: 1.25 μm). We derived the average reflectance in two encircled locations with 8 km diameter: (A) inside the crater and (B) at some distance. **b** Reflectance spectra of the areas A and B and their ratio. The phase angle of observation ($\alpha = 33°$) was the same at both locations within 1°. Data below and above 1.05 μm were acquired with separate detectors (VIS and IR). The IR spectrum was calibrated with corrections from ref. [54]. Remaining calibration artifacts, visible as yellow and cyan coloured lines in the colour composite and narrow spikes in the spectra, do not affect the spectral slope.

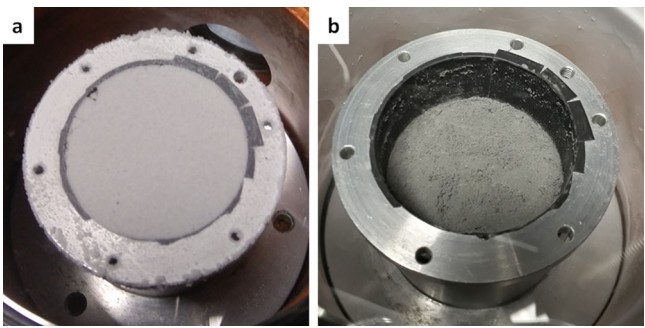

**Fig. 3 The experimental sample seen through the sapphire window. a** The container filled with the mixture of ice and Ceres analogue material at the start of the sublimation experiment. **b** The same container with the sublimation residue after 133 h under high vacuum at 173 K.

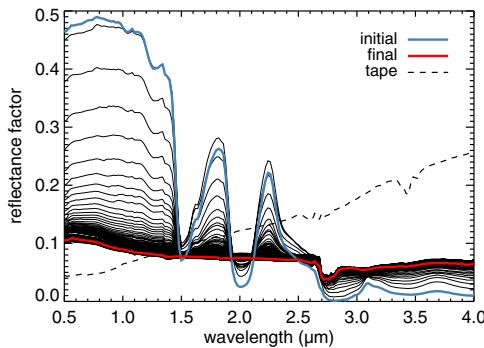

**Fig. 4 Spectral changes during the Ceres analogue sublimation experiment.** Spectra of the ice sample under vacuum at 173 K were acquired in hourly intervals at the standard geometry of $(\iota, \epsilon, \alpha) = (0°, 30°, 30°)$. The first and last spectrum in the series are highlighted in blue and red, respectively. The initial spectrum is that recorded at the start of the experiment under medium vacuum. The final spectrum is the last one of the series recorded after 133 h under high vacuum. The dashed line is the spectrum of the empty container, which is coated with black aluminium tape.

## Results

**Sublimation phase.** At the start of the sublimation experiment, the icy sample of the Ceres analogue material appeared pale grey to the eye (Fig. 3a). In the following days, the surface of the sample receded towards the bottom of the container and

developed what looked like a crust, dotted with tiny (mm-sized) sublimation pits. After 6 days the sample had descended all the way towards the bottom of the container, yet the black aluminium tape at the bottom of the container was not visible through the residue. After opening the chamber window, we removed the container for physical and mechanical characterization of the residue. The residue surface was flat and pitted (Fig. 3b). It appeared grey to the eye and brighter than the original Ceres analogue material. The residue thickness was about 1 mm, and no water ice was found to remain below.

Figure 4 shows all SHINE spectra acquired from the beginning to the end of the 133 h long sublimation experiment. The first reflectance spectrum (blue curve) is that of the icy sample under medium vacuum. The reflectance of the sample is expected to be stable under these conditions, as they induce only minimal sublimation. The spectrum shows strong water ice absorption bands at 1.5 and 2.0 μm. After applying high vacuum, these bands disappeared while the ice sublimated and were less than 10% of their original depth after 24 h. During the 50-min spectral acquisition, the sample would evolve. The spectrometer would record the resulting spectral changes as it progressed along the wavelengths. The depth of the ice bands would typically decrease during the acquisition, leading to an asymmetric appearance of the absorption bands. Occasional small jumps in reflectance over the entire wavelength range probably resulted from changes in the surface structure in the SHINE field of view, like the appearance of the aforementioned pits. The last spectra in the series are virtually identical and show no evidence of water ice bands (red curve in Fig. 4). Since the residue was so thin (1 mm), the last spectra may be affected by reflected light from the black aluminium tape just below. The reflectance spectrum of the tape (dashed curve in Fig. 4) has a red slope, which makes the tape more reflective than the residue beyond 1.3 μm. It also features absorption bands around 2.7 and 3.5 μm. The absence of these bands from the reflectance spectrum of the residue implies that the contribution of the tape is minor. If it does contribute to some extent, it makes the residue appear redder than it really is. The absence of a clear spectral contribution of the black aluminium tape demonstrates that reflected light mostly originates in the top 1 mm of the residue surface.

**Residue properties.** The sublimation residue had greatly increased in volume compared to the original material (estimated by a factor 5–10), and was lightweight with a foam-like texture (Fig. 5a). It was easily compressed, yet flexible with some internal

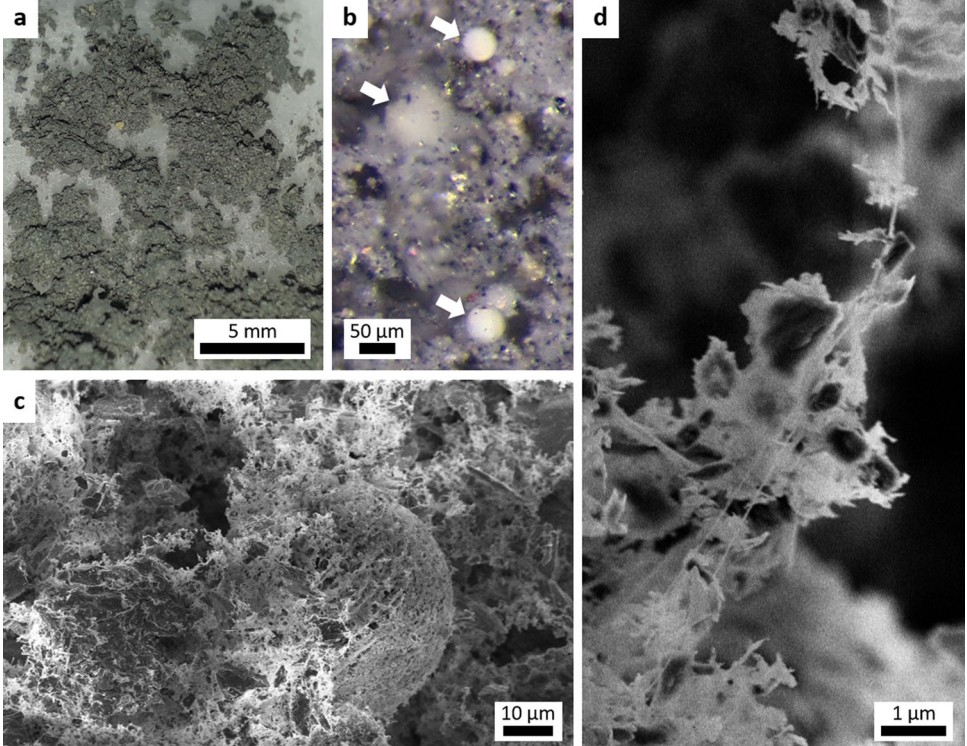

**Fig. 5 Micrographs of the Ceres analogue sublimation residue. a** Close-up photo of the residue on cotton wool. **b** OM image showing pale spheres (arrows) embedded in a grey matrix. **c** SEM image revealing the high porosity of the residue. **d** SEM close-up of a filament.

cohesion (higher tensile strength than compressive strength); we cut it to obtain fragments. We characterized the residue with an optical microscope (OM) and scanning electron microscope (SEM). The OM image (Fig. 5b) shows pale spheres of about 50 µm size on a grey background, in which we can distinguish small, black particles (mostly magnetite) and larger, yellowish particles or concretions. We identify these spheres as former ice particles. They appear bright because they are exclusively composed of nontronite (only the nontronite was incorporated into the ice particles, see section 'Sample preparation'). The grey material incorporated the other mineral particles that were mixed in with the ice. The SEM images in Fig. 5c and Supplementary Figs. 7–10 show the residue in greater detail. It is entirely porous and is composed of a network of filaments with sub-µm structure (Fig. 5d). The spheres can be distinguished from the background material only by their roughly spherical outline, suggesting the latter originates from broken spheres. Compact particles are found scattered throughout the network, but only outside the spheres. SEM images of the nontronite-only sublimation residue show a similar network of spheres and sphere fragments that appears to be even more porous (Supplementary Figs. 12 and 13). The extremely porous structure of the residues was also noted in earlier sublimation experiments with smectite phyllosilicates[20,21].

Micrographs in Fig. 6 show compositional maps of the residue that reveal the fate of the minerals in the original material. The energy-dispersive X-ray spectroscopy coupled with the SEM allows mapping the distribution of several elements. The SEM images (Fig. 6a, b) show that the original material is composed of angular particles, whereas the sublimation residue is composed of angular particles and spherical objects set in a porous matrix. The elemental maps (Fig. 6c, d) show the distribution of Fe (red), Ca (green), and Si (blue). Fe mostly traces magnetite, which particles appear bright in the backscattered SEM image. Magnetite was not dispersed and is present as angular particles in both the original material and sublimation residue. Ca traces dolomite, which is

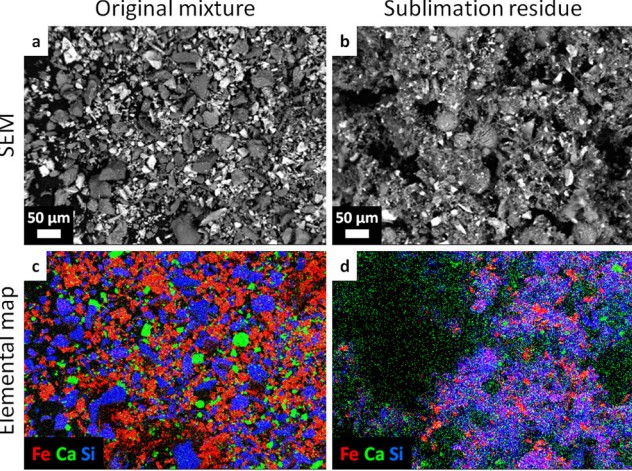

**Fig. 6 Mineral distribution in the Ceres analogue sample, pre-and post-sublimation.** SEM (backscattered electrons) images (**a**, **b**) and corresponding elemental maps (**c**, **d**) of the original Ceres analogue material (**a**, **c**) and the sublimation residue (**b**, **d**). The elements Fe, Ca, and Si are mapped in red, green, and blue, respectively. The 3D-structure of the residue leads to some parts of the scene in **b** not registering in the elemental map **d**.

found dispersed as fine particles in the sublimation residue. Si traces antigorite and nontronite. The phyllosilicate particles completely dispersed during the experiment and their material makes up most of the porous network in the residue, including the spheres. Whereas the surface in Fig. 6a, c) is a flat, thin layer of solid particles, the surface in Fig. 6b, d is a porous, three-dimensional structure. The elemental map (Fig. 6d) has little depth and shows only the top layer of the residue; the deeper layers seen at left in Fig. 6b are not recognized in Fig. 6d. Because

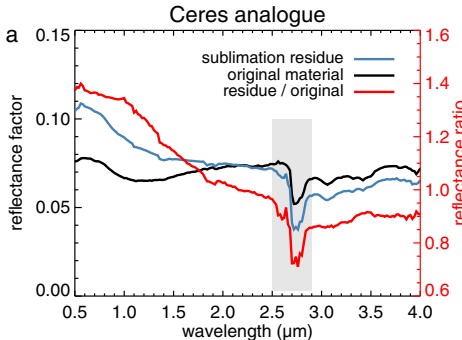
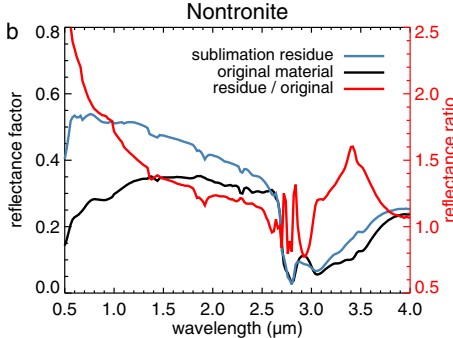

**Fig. 7 Spectral changes induced by the sublimation process at the standard geometry. a** Ceres analogue sample. The reflectance of the sublimation residue features absorption bands associated with water vapour, which affects the band depth in the range indicated by the grey box. **b** Pure nontronite sample (Supplementary Note 1). The ratio spectra are characterized by negative spectral slopes up to about 2.5 μm, meaning that the sublimation residues are bluer than the original material.

the volume of the nontronite foam in the residue is so much larger than that of the original nontronite particles, the number density of magnetite particles (red) is much lower in Fig. 6d than in Fig. 6c.

**Spectral changes**. The sublimation experiment led to changes in the spectrum of the Ceres analogue material. Figure 7a shows the last spectrum acquired during the sublimation experiment (blue) as well as the spectrum of the original material (black) and their ratio (red). In the visible range, the sublimation residue is about 40% brighter than the original material. The ratio spectrum has a negative slope over the visible and near-IR wavelength range, meaning that the residue is bluer than the original material. The sublimation residue of pure nontronite is also brighter and bluer than the original powder (Fig. 7b), which suggest a critical role for the foamy structure. The brightening and bluing of the residues may be due to the presence of abundant scattering centres in the porous nontronite network that are small enough to scatter in the Rayleigh regime[20,26]. For Rayleigh scattering to occur, the scattering centres need to be smaller than the wavelength ($\ll$1 μm) and scatter independently, i.e., be separated by a distance larger than the wavelength[24,25]. The filaments in Fig. 5d appear to be small enough (<200 nm), and the high porosity of the nontronite foam naturally provides the necessary distance between the scattering centres (Fig. 5c). In this way, the nontronite foam is perhaps similar to aerogel, which blue appearance is attributed to Rayleigh scattering[27]. The bluing is less strong for the Ceres analogue residue than the nontronite residue, which is probably related to the presence of the other mineral particles. Beyond 3 μm, the spectral shape is similar for both materials, with the residue being 10% darker, perhaps because of the presence of pits on the surface. The 2.7 μm absorption band is pronounced in the ratio spectrum, indicating that it is deeper for the residue. However, the residue spectrum is affected by absorption bands associated with water vapour present in the optical path (Supplementary Note 3), which renders the depth of the 2.7 μm band uncertain. The montmorillonite residue studied by Poch et al.[20] had spectra with hardly detectable absorption bands of adsorbed water and OH groups that were clearly present in the spectrum of the original montmorillonite.

Comparing the spectral changes observed in our experiment with those in Haulani crater on Ceres (Fig. 2), we find that they are similar in character. In both cases, the reflectance has increased at visible wavelengths and the reflectance ratio spectrum features a negative slope from the visible to the near-IR up to 3 μm (bluing). The experimental slope is steeper by about a factor two. Knowing that bluing lessens over time on Ceres[12], this may be related to the

fact that the experimental residue is fresh, whereas Haulani's age is estimated to be at least 1 Ma[12,14]. Other possible reasons are the lower albedo of the Haulani blue material and differences in the ice-to-phyllosilicate ratio.

## Discussion
On Ceres, any sufficiently large impact would intimately mix the phyllosilicates present on the surface[5,7] with the water ice present just metres below[8–10] and expose this muddy suspension on the surface, at least partly in a liquid state[28]. Flow features associated with craters like Occator and Haulani have been interpreted as impact melt involving liquid water[29,30]. In a recent experiment under low-pressure conditions, a frozen crust quickly formed on the surface of a flowing, low viscosity mud by evaporative cooling, akin to basaltic lavas flows on Earth[31]. Vigorous boiling was observed, which did not impede formation of the crust. As crust formation should be even more efficient in vacuum, the surface of the ejecta of fresh craters on Ceres is expected to freeze quickly[31]. This provides the opportunity for sublimation and the formation of a porous residue. The mud immediately below the crust will freeze at leisure, providing a reservoir for later sublimation after removal of the crust by small meteorite impacts. The time it takes to sublimate ice from the surface down to about a metre depth has been estimated at about 2 Ma at the equator and 20 Ma at 40° latitude, assuming a porosity of 0.5 (ref. [18]). The residue in our experiment was extremely porous, as experimental constraints limited the concentration of phyllosilicate in the ice to 1% by weight. On Ceres, the residue may be of higher density and thereby less porous, but still offer increased (vacuum) insulation through its low thermal conductivity[21]. Depending on the thickness of the porous layer, water ice could be stable at lower depths on Ceres than previously assumed.

Our experiment suggests that sub-μm-sized filaments of phyllosilicate composition inside the residue scatter light in a similarly blue fashion as observed on Ceres, with the foam-like structure providing the necessary distance between the scattering filaments for Rayleigh scattering to occur[24,25]. A porous network is required, because a surface made of sub-μm-sized particles in close contact would merely reflect like a slab[25]. Apart from their blue colour and higher reflectance, there is some evidence that the ejecta of fresh craters are in a different physical state than their surroundings. The blue ejecta of Azacca crater possess an unusually narrow opposition effect[32]. If it is due to shadow hiding, expected for a material this dark, Hapke's theory predicts a high porosity[33]. A similarly narrow opposition peak for the moon Europa was interpreted in terms of a very high porosity, assuming shadow hiding[34].

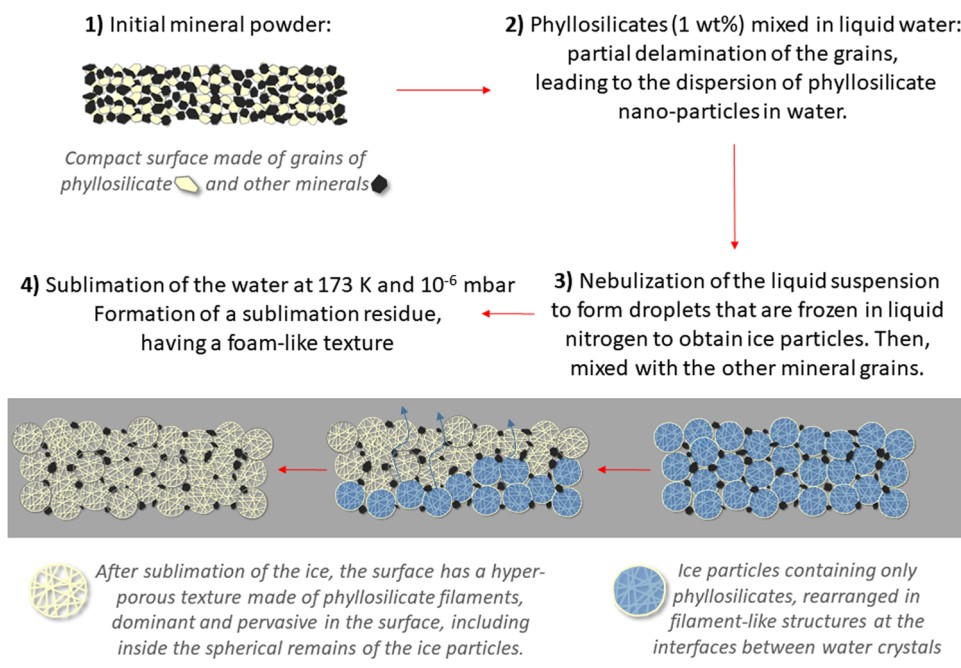

**1) Initial mineral powder:**

*Compact surface made of grains of phyllosilicate ◯ and other minerals ●*

**2) Phyllosilicates (1 wt%) mixed in liquid water:** partial delamination of the grains, leading to the dispersion of phyllosilicate nano-particles in water.

**4) Sublimation of the water at 173 K and 10⁻⁶ mbar** Formation of a sublimation residue, having a foam-like texture

**3) Nebulization of the liquid suspension** to form droplets that are frozen in liquid nitrogen to obtain ice particles. Then, mixed with the other mineral grains.

*After sublimation of the ice, the surface has a hyper-porous texture made of phyllosilicate filaments, dominant and pervasive in the surface, including inside the spherical remains of the ice particles.*

*Ice particles containing only phyllosilicates, rearranged in filament-like structures at the interfaces between water crystals*

**Fig. 8 Diagram showing the sublimation process for the Ceres analogue material.** Inside each spherical ice particle produced through the experimental SPIPA-B protocol, phyllosilicate platelets formed veins between the water ice crystals. During the sublimation of the ice, the water crystals were replaced by voids and the platelets remained to form a porous network. In some cases, the desiccated structure retained the spherical shape of the original ice particle. More often, spheres fragmented into a disordered fluffy medium of platelets and filaments.

The space environment will erode the sublimation residue over time, leading to the observed decrease of the blue spectral slope of the Ceres spectrum over millions of years[12]. Our experimental results suggest that the reflected light mostly originates in the upper 1 mm of the residue surface. How would space weathering affect a top layer of porous phyllosilicate? The Ceres analogue residue was flexible and had an internal cohesion that made it resilient[21]. The mechanical properties of the much thicker non-tronite residue layer were easier to study, and we found it to be cohesive, sticky, and spongy (Supplementary Note 1). The residue would be insensitive to thermal stresses, so erosion would mostly result from particle impact. Space weathering by particles has been studied extensively for lunar rocks returned by Apollo project. Energetic particles leave tracks in the solid rock, whereas micrometeorites create mm-sized pits on the surface[35,36]. The top few mm are dominated by tracks caused by alpha particles and protons from solar flares, whereas heavy nuclei from galactic cosmic rays leave tracks at a depth of more than 5 mm. The action of these particles is expected to erode lunar rocks by ~1 mm per million years[35]. It is not clear how energetic, charged particles would interact with the porous sublimation residue. Solar particles may gradually break down the phyllosilicate structure, also because the introduction of charge may affect electrostatic and ionic bonding forces. However, they will be less effective than on the Moon due to the larger distance of Ceres from the Sun. Cosmic rays might essentially be absorbed by the residue, travelling deep into the interior before depositing their energy. Likewise, compact micrometeorites might also barely interact with the top layer of the residue; the residue's low compressive strength will allow such particles to punch right through and only gradually release kinetic energy on their way down, akin to the penetration tracks in the aerogel carried by the Stardust mission[37]. If they reach as far down as the ice layer, any resulting mini-explosions may expose fresh muddy ice and refresh the residue surface rather than erode it. Small, very blue craters on the blue ejecta of Ikapati crater demonstrate this process on a larger scale[29]. We can only speculate on the exact

nature of the fading of the blue colour. Other possibilities are alteration of the physical structure or chemistry of the blue material by energetic particles, or mixing with underlying or nearby regolith by impacts[38,39].

The origin that we propose for the blue colour is not the only one possible. An alternative explanation holds that phyllosilicate aggregates spontaneously form on the ejecta of fresh craters, providing the spacing between the scattering centres that is necessary for Rayleigh scattering[13]. But, if impacts pulverize phyllosilicates to produce large quantities of very fine particles, aggregates would not form spontaneously. Structures called fairy castles, for example, are formed by gently sprinkling the powder[40,41]. If micrometeorites can produce such aggregates, we would expect all of Ceres to be blue. It is also not clear how aggregates would form in muddy ejecta[29,30], if not through the mechanism that we propose.

In conclusion, we argue that a porous network of phyllosilicate filaments is responsible for the blue colour of fresh craters on Ceres. Our experiment provides support for a natural origin for such structure: Impacts produce a phyllosilicate slurry that quickly freezes on the surface and gradually sublimates, leading to the formation of a porous residue. The residue provides a network of sub-µm-sized scattering centres, whose spacing is sufficiently wide for Rayleigh scattering to occur. The blue colour will fade over time as the network is gradually destroyed by space weathering.

## Methods

**Experimental setup**. The experiment was performed in April 2019 at the Institut de Planétologie et d'Astrophysique de Grenoble in Grenoble, France, using the Cold Surface Spectroscopy (CSS) facility (https://cold-spectro.sshade.eu/). We produced and sublimated ice particles by following the protocol of Poch et al.[20]. The experimental procedure is outlined in Fig. 8. The first step was to suspend the analogue material in water and produce ice particles with help of liquid nitrogen. Keeping this icy sample in a cryogenic chamber under conditions of low pressure and temperature over the course of several days resulted in the formation of a sublimation residue. A spectro-gonio radiometer monitored the residue during the experiment. After opening the chamber at the end of the experiment, we

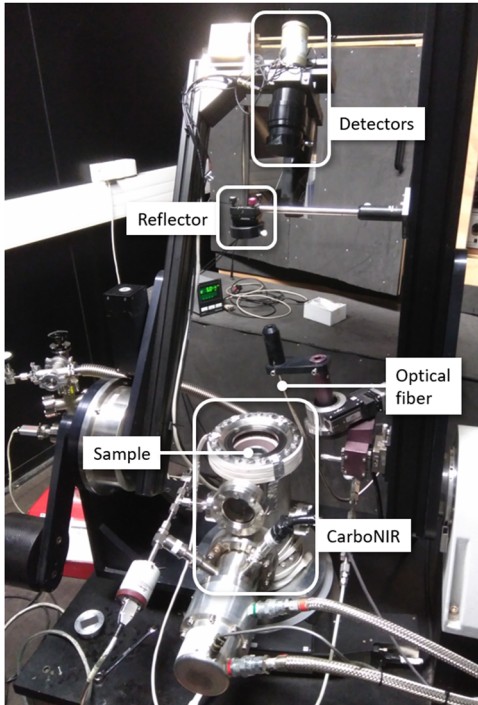

**Fig. 9 Setup of the sublimation experiment: The SHINE spectro-gonio radiometer with the icy sample inside the CarboNIR environmental chamber.** Light originating in a monochromator (not visible) is led through an optical fibre to a reflector to illuminate the sample at 0° incidence angle. Detectors at the end of an arm of SHINE captures light reflected by the sample at an angle of 30°.

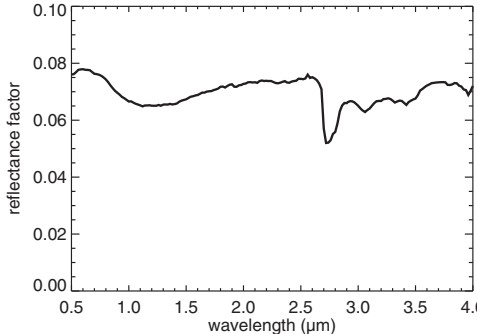

**Fig. 10 Reflectance spectrum of the Ceres analogue material acquired at the standard geometry.** The measurement uncertainty (standard deviation of 10 repeats) is smaller than the width of the drawn line over the entire range.

characterized the residue morphologically and recorded its mechanical properties. We performed the experiment twice: first with pure nontronite and then with Ceres analogue material. Here we report on the results of the Ceres analogue experiment, as the pure nontronite experiment was primarily performed to practice the procedure. Nevertheless, the nontronite sublimation residue had relevant structural, mechanical, and optical properties that we will occasionally refer to. The nontronite experiment is described in Supplementary Note 1.

The experimental setup with spectro-gonio radiometer and cryogenic chamber is shown in Fig. 9a. The icy sample was held by a cylindrical container of 4.8 cm diameter and 2.0 cm depth, which had been coated with black matte aluminium tape on the inside to limit secondary reflections. The container was kept inside CarboNIR, a cylindrical environmental chamber designed for analysing surfaces at low temperature[42]. The copper chamber has high thermal conductivity and can be cooled by a cryostat to stabilized temperatures down to 77 K. CarboNIR is located in the centre of a larger stainless steel chamber that is connected to a vacuum pump. The chamber can either be put under medium vacuum (~0.14 mbar of gaseous nitrogen) or high vacuum (~$1.5 \times 10^{-6}$ mbar). It has a removable lid on top with a transparent, sapphire window, through which the reflectance of the sample was measured by the spectro-gonio radiometer SHINE (SpectrophPotometer with variable INcidence and Emergence)[43]. The instrument operated in the special Gognito mode, optimized for dark targets, in which the sample illumination and observation areas are disks with a diameter of 7.1 and 20 mm, respectively[44]. The monochromatic light used for illuminating the sample is modulated and the detection synchronous. Thereby, SHINE measures light reflected by the sample only, and is neither sensitive to scattered light from external sources nor thermal emission. The accuracy of its absolute radiometric calibration is better than 0.1% over the 0.4–2.5 μm range. The spectra were recorded over the 0.4–4.2 μm range with a fixed sampling of 20 nm and a spectral resolution of 3.2 nm below 0.68 μm, 6.3 nm below 1.4 μm, 13 nm below 3.0 μm, and 26 nm above 3.0 μm. For each wavelength interval, the intensity was recorded by a 300 ms integration repeated 10 times. The calibrated reflectance factor was derived from the average of these 10 values, and their standard deviation (typically <0.5%) was adopted as the associated uncertainty. The time needed to acquire a single spectrum was about 50 min, during which time the spectrometer progressed along the wavelengths. Prior measurements of reference surfaces allowed calibration of the measurements to reflectance factor (Supplementary Note 2). All measurements in this experiment were performed at the standard geometry of $(\iota, \epsilon, \alpha) = (0°, 30°, 30°)$, with incidence angle $\iota$, emission angle $\epsilon$, and phase angle $\alpha$. For this geometry, the reflectance factor (REFF) is identical to the radiance factor (I/F). The presence of the sapphire window over the sample induces secondary reflections between

sample and window, which reduces the measured reflectance and contrast of absorption features. The calibration includes a correction for this effect[45]. We verified the accuracy of the SHINE spectra of the original sample and sublimation residue by repeating the measurements with SHADOWS (Spectrophotometer with cHanging Angles for the Detection Of Weak Signals), which is a similar spectro-gonio radiometer optimized for dark surfaces[44].

**Sample preparation**. The Ceres analogue material was produced as a fine powder (grain size <36 μm) at the Istituto di Astrofisica e Planetologia Spaziali-INAF in Rome, Italy. This analogue material was a mixture composed of 44 vol% (63 wt%) of magnetite ($Fe_3O_4$), 35 vol% (22 wt%) of $NH_4$-bearing nontronite, 13 vol% (9 wt%) of antigorite (($Mg$, $Fe$)$_3Si_2O_5(OH)_4$), and 8 vol% (6 wt%) of dolomite ($CaMg(CO_3)_2$). The $NH_4$-bearing nontronite was produced according to Ferrari et al.[6,46], starting from the $NH_4$-free nontronite (NAu-1)[47]. SEM images of both samples are provided in Supplementary Figs. 6 and 11. The analogue material appeared very dark to the eye and was about twice as bright as the average Ceres surface[3], with a reflectance of around 0.07 at the standard geometry (Fig. 10). After spending several hours in an oven at 90 °C under medium vacuum to remove the most labile water molecules adsorbed to the phyllosilicates, SHINE acquired a spectrum of the analogue material in the CarboNIR chamber under high vacuum and room temperature.

We aimed to make an icy intra-mixture, in which all components of the analogue material are present as inclusions in the water ice grains (as opposed to an inter-mixture, in which the material particles are mixed with pure water ice particles)[20], and followed the protocol and setup for production of icy planetary analogues as described and reviewed by Pommerol et al.[48]. On Tuesday 23 April, we added 0.581 g material to 58.5 ml ultra-pure water in a glass beaker on a magnetic stirrer to achieve a 1% suspension by weight. We chose a 1 wt% concentration for practical reasons; it had been proven to work[20], and higher concentrations would run the risk of clogging the tubes and the sonotrode. The analogue material was not easily dispersed, and we resorted to ultrasound to disrupt and suspend floating aggregate particles. Then, we led the suspensions through an ultrasonic nebulizer equipped with a sonotrode, which produced a spray of fine droplets. The spray was aimed towards liquid nitrogen in a container that had been placed inside a large freezer. The freezer was kept at −30 °C and also held the tools used for manipulating the ice sample, immersed in liquid nitrogen at all times. Contact of the spray with the liquid nitrogen resulted in the formation of icy particles. These particles, produced according to the SPIPA-B protocol, are ice spheres with a diameter of 67 ± 31 μm[20,49,50]. After emptying the beaker of most of the Ceres analogue suspension, a dark, particulate residue was left behind. We dried the residue in an oven under vacuum at 90 °C and momentarily kept the powder apart. Furthermore, magnetite particles had attached themselves to the rotating magnet of the stirrer. After drying, we carefully separated the magnetite from the stirrer magnet with the help of a stronger magnet behind a piece of paper. We then added the magnetite particles to the other residual powder. The mineral residue (now including 81 wt% magnetite) was weighed at 0.468 g, meaning that only ~20 wt% of the analogue material had been incorporated in the ice particles. The similarity of this number to the abundance of nontronite in the original sample (22 wt%) suggests that only nontronite was incorporated into the ice. We mixed the residue with the ice particles in an aluminium bottle, pre-cooled with liquid nitrogen, using a vortex shaker. In all, 0.178 g of residue was combined with 22.31 g of ice particles to meet our original goal of a 1 wt% solid-to-ice mixture. Thus, the final analogue ice sample was a combination of an intra- and inter-mixture, having the same mineral weight proportions as the original analogue material. Inside the freezer, we sieved the ice mixture (mesh size 400 μm) to fill the 2-cm-deep sample holder to the rim, smoothing the surface with a spatula. The holder and spatula had been pre-cooled with liquid nitrogen. A spectrum of the empty sample holder had been acquired at 173 K and high vacuum to verify that the sample was optically

thick over the course of the experiment. The full sample holder was quickly transferred from the freezer to the CarboNIR cryogenic chamber inside a polystyrene box cooled with liquid nitrogen.

**Sublimation experiment**. We put the icy sample in the CarboNIR chamber around 18:00 h on Tuesday 23 April. Initially, we held it under medium vacuum in order to record a spectrum of the mixture under stable conditions. We proceeded to high vacuum after about 2 h to start the sublimation of the ice. The chamber interior was kept at a temperature of $173 \pm 1$ K throughout the experiment, which is in the range of the mean surface temperatures experienced on Ceres at lower latitudes[18]. During the experiment, SHINE acquired one spectrum every hour and we visually inspected the sample through the window twice per day. The first and last spectrum in the series were acquired on 23 April, 19:40 h and 29 April, 08:40 h, respectively. After 6 days, the hourly spectra showed no perceptible changes and the ice absorption bands had disappeared completely. We inferred the absence of water ice below the surface from the duration of the sublimation experiment (by comparison with previous sublimation experiments) and the absence of any motions on the surface of the residue caused by the passing flow of water vapour. We removed the sample from the CarboNIR chamber while it was still at 173 K. To keep it cold and dry, we temporarily kept it in a box with liquid nitrogen for an initial characterization. We measured the sample thickness to be slightly less than 1 mm using a ruler. When lifting the dust mantle, we found no water ice below. The sample was then warmed to room temperature for further characterization.

## Data availability

All laboratory spectra are available in the Cold Surfaces Spectroscopy Facility (CSS) on SSHADE[51,52]. Global colour maps of Ceres used in Fig. 1 that were derived from Dawn framing camera images are available at https://doi.org/10.5281/zenodo.4251217. Dawn VIR calibrated (level 1b) data used in Fig. 2 are from the HAMO orbit (cycle 2) and are available at https://sbn.psi.edu/pds/resource/dawn/dwncvirL1.html.

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

## Acknowledgements

The experimental work was performed in the Europlanet 2020 RI framework, which has received funding from the European Union's Horizon 2020 research and innovation programme under grant agreement No 654208. O.P. acknowledges a postdoctoral fellowship from the Centre National d'Études Spatiales (CNES) for funding. S.M.P. is supported by Université Grenoble Alpes (IRS IDEX/UGA). P.B. and O.P. acknowledge funding from the European Research Council under grant SOLARYS (77169). This work has been supported by the Italian Space Agency (ASI) through grant ASI I/004/12/2. We thank Frédéric Charlot for assisting with the SEM.

## Author contributions

S.E.S. analysed the data and prepared the manuscript. O.P., S.E.S., and R.S. performed the experiment, with S.M.P. providing support. S.D.A. and M.F. prepared the samples. R.S. acquired SEM images. P.B., M.C.D.S., B.S., and all other authors reviewed the manuscript and provided comments and input at all stages.

## Funding

## Competing interests

The authors declare no competing interests.
