## [Peer Review File · Nature Communications]

REVIEWER COMMENTS

Reviewer #1 (Remarks to the Author):

Review of

Bluing on Ceres as a Natural Consequence of Sublimation

by Schröder et al.

Review by Roger Clark. I do not wish to remain anonymous.

This is an interesting paper that brings new light onto the question of observed bluing of surfaces in the Solar System and Ceres specifically. As seen below, the paper requires a few issues to be addressed before publication.

Page 2:

The authors say (lines 47-48) that "Rayleigh scattering inside a particulate surface has not yet been adequately modelled." But Clark et al., 2010 (reference below) did just that, both analogs and radiative transfer modeling. Brown (2014) developed the problem further.

Lines 126-128:

On Tuesday 23 April, we added 0.581 g material to 58.5 mg ultra-pure water in a glass beaker on a magnetic stirrer to achieve a 1% suspension by weight.

0.581 g material to 58.5 mg would be water/material = 10% water

Do you mean 58.5 grams, not mg?

Page 6 lines 236-239:

"Sublimation introduced more intense bluing for the nontronite than for the analogue material, which suggests that the phyllosilicate foam is predominantly responsible for the blue colour, perhaps through Rayleigh scattering [5, 27]."

Bluing is caused by sub-micron scattering, e.g. sub micron particles in an intimate mixture (Clark et al., 2010).

So the experiment only suggests that dispersal of fine particles or sub-micron scattering centers is responsible, not necessarily that sublimation is responsible. Such scattering, for example, could be created by micrometeoroid bombardment, as previously noted in the manuscript.

Page 7, lines 297-298:

"Likewise, micrometeorites might also barely interact with the top layer of the residue."

Why? A micrometeorite will "interact" with the surface on down until all the energy is deposited. To get to the subsurface, the upper surface layers must be penetrated, thus broken up. And breaking up the surface, liberating embedded fine particles would increase the scattering, thus increase bluing.

Page 7, lines 304+

The experiment does not prove "the leading candidate responsible for the blue color of fresh craters on Ceres is phyllosilicate structure on a sub- μ m-scale." It only gives another possible explanation.

I refer the authors to:

Clark, R. N., D. P. Cruikshank, R. Jaumann, R. H. Brown, K. Stephan, C. M. Dalle Ore, K. E. Livo, N. Pearson, J. M. Curchin, T. M. Hoefen, B. J. Buratti, G. Filacchione, K. H. Baines, and P. D. Nicholson, 2012, The Composition of Iapetus: Mapping Results from Cassini VIMS, *Icarus*, 218, 831-860.

There modeling and analog experiments were done to better understand the conditions where particles scatter vs absorb. Bluening and Rayleigh scattering is a diffraction phenomenon. The results in Clark et al show that bluing occurs when a tiny particles with a different index of refraction is separated from the host matrix (an intimate mixture like salt and pepper). If the different index of refraction particle is embedded in another matrix so that the index of refraction difference is smaller, scattering is decreased, e.g. pepper inside a salt grain, then the particle can absorb more than scatter. In space weathering we see this effect: if the iron particles are separated from the matrix (salt and paper mixture), we see bluing (see Figure 17 of Clark et al, and models in Figure 21). Embedded particles show absorption, e.g. see Clark et al, Figures 23a, 23b. Further, the bluing occurs when they

scattering centers are widely separated. If the sub-micron particles become too dense, then scattering by diffraction is no different than scattering by larger particles and the surface reflectance begins to look like a slab. For example, 10 micron grains vs 1-micron grains, vs 1 nm grains in contact. Visible and UV photons will not see 1 nm grains any different than a solid slab of the same material.

The authors claim that the bluing is caused by "phyllosilicate structure on a sub- μm -scale." However they do not show high enough magnification to evaluate that. The magnification needs to be high enough to show sub 200 nm detail, and better yet, sub 50 nm detail.

For example, add to figure 8, images with 1 micron scale bar that is larger than the 50 micron scale bar in the present figure.

Sub-micron phyllosilicate particles embedded in a salt matrix will have a small index of refraction difference and show absorption more than scattering. It would be sub-micron particles outside the matrix that predominantly contribute to bluing. Further, salt along with sub-micron structure will not contribute significantly to bluing; it will mostly brighten the surface. Complication of this, which has not been previously studied is the effect of porosity on bluing. This is another area where the authors can contribute significantly to understanding this phenomenon.

I suggest that the authors address these issues and resubmit. It is a good paper that will not take much work to make it much better.

Roger Clark

Reviewer #2 (Remarks to the Author):

Bluing on Ceres as a Natural Consequence of Sublimation

Reviewer: Kerri Donaldson Hanna, University of Central Florida

I find this manuscript to be a highly relevant and timely piece of work that should be published.

What are the major claims of the paper?

The manuscript describes a physical mechanism, one that is expected to happen on the surface of Ceres, for the observed spectral 'bluing' effect observed at young craters. The proposed mechanism is that impacts into Ceres' surface creates a phyllosilicate-rich slurry that freezes and then gradually sublimates, leading to the formation of a porous residue that scatters light in a preferential way.

Are they novel and will they be of interest to others in the community and the wider field? If the conclusions are not original, it would be helpful if you could provide relevant references.

The manuscript describes a set of experiments and laboratory spectra that demonstrate this spectral bluing as ice sublimates from a phyllosilicate and ice mixture held at Ceres conditions. In addition, the original starting material and residue left over from the sublimation experiments are investigated with scanning electron microscopy to show the porous nature of the residue materials and understand how the original material evolves during the sublimation process.

Is the work convincing, and if not, what further evidence would be required to strengthen the conclusions?

The work does not require any further evidence to strengthen the conclusions. I do suggest adding error bars to at least of the spectral measurements (see comments below) to demonstrate the expected variability for measurements taken over a 50 minute integration time.

On a more subjective note, do you feel that the paper will influence thinking in the field? Please feel free to raise any further questions and concerns about the paper.

Yes, the paper will influence not only the observed spectral differences on Ceres, but also on other bodies where you have ice and rock interacting during the impact process.

We would also be grateful if you could comment on the appropriateness and validity of any statistical analysis, as well the ability of a researcher to reproduce the work, given the level of detail provided.

The manuscript and supplementary materials provide a well-described methodology for the laboratory experiments and the spectral measurements. The nature of these laboratory experiments are quite complex and require bespoke facilities for creating the analogue-ice mixtures, holding the samples at Ceres-like conditions, and collecting spectral measurements over time. Thus, it would be quite difficult for these experiments to be replicated elsewhere. However, I do think the manuscript provides the necessary discussion to understand the experimental set-up and laboratory spectra. While the scope of the manuscript is not to quantify the depths of any of the spectral bands, it would be helpful to include a figure or discussion on the variability of spectral signature over the ~50 minutes needed for each measurement. Adding error bars to one of the spectra, showing that variation would be helpful in demonstrating any changes with time (if any).

Reviewer #3 (Remarks to the Author):

Comments on “Bluing on Ceres as a natural consequence of sublimation” by S. Schroder et al:

Reviewer: Bruce Hapke

This paper presents the results of laboratory experiments intended to support the hypothesis that the blue color of features in and around new impact craters on Ceres is a property of the residue left from sublimation of a frozen slurry of regolith and water ice produced by the impact. The paper is timely, interesting, clearly written and will be of interest to planetary scientists and others. However, the experimental results presented are ambiguous and there is an alternate interpretation that the paper does not consider. I cannot recommend publication of the paper in its present form. The

authors need to discuss other explanations and make additional measurements that would resolve this ambiguity and also address the other comments below.

The model for making the blue color on Ceres postulates a large impact into a mixture of water ice and regolith located below the surface. The impact melts the ice, producing a slurry of regolith and liquid water, some of which is ejected and deposited outside the crater and some remains on the inside surface of the crater. The water freezes and then slowly sublimates, leaving a blue residue.

The experiment tests this hypothesis by using a nebulizer to spray droplets of a 1% slurry of Ceres regolith simulant, consisting of magnetite, nontronite and other phyllosilicates, and water onto liquid nitrogen. The resulting frozen droplets are mixed with a residue that had settled out from the slurry during the spraying and remained in the beaker feeding the nebulizer. The mixture was placed in a vacuum and kept at 173K for several days until the ice had completely sublimated. The sample remaining was blue in color. Electron microscope images revealed that it contained large numbers of sub-micron filaments, believed to be dissolved and precipitated phyllosilicate material, and it is hypothesized that Rayleigh scattering by these structures causes the blue color. There are several problems with the model and the experiments that should be addressed before the paper is accepted:

(1) The Ceres regolith simulant is chemically unrealistic. Both magnetite and nontronite contain ferric iron, but conditions on Ceres are not expected to be oxidizing, so neither mineral is likely to occur there.

(2) The experiment does not realistically simulate conditions on Ceres. In the experiments the slurry remained in liquid form for some time while being disaggregated and nebulized. However, when liquid water is exposed to a vacuum it does not just sit quietly, but evaporates explosively. On Ceres the slurry would boil explosively as soon as it is melted during the impact, rapidly dispersing the solid components in all directions and preventing the formation of the submicron structures. A thick deposit of slurry might remain liquid for a short time while running downhill and could form lobate structures, but it would be boiling violently the whole time.

(3) Figure 9 shows the spectra of the original Ceres regolith simulant and the sample remaining after the sublimation process. In the visible the sample has a high reflectance and a steep negative spectral slope, which causes the blue color, while the simulant has a low reflectance with a much smaller negative slope, and is dark. Inspection of the spectra suggests that the simulant spectrum can be made by adding a dark substance with a relatively flat spectral reflectance to the sublimed sample. Magnetite is the obvious candidate for this substance.

Conversely, this inference means that the treatment the simulant received, suspension in water, disaggregation, sedimentation, nebulizing, freezing and subliming, somehow resulted in the selective loss of magnetite relative to the other constituents. (For example, lines 138-143 state that magnetite particles were magnetically separated from something, but it is not clear what that something is. Was it the stirrer magnet, the beaker residue, or what? Why was this done? What was done with this magnetite? Was it mixed back in with the residue, discarded, or what?)

This inference is supported by figure 8; magnetite is conspicuous by its abundance in the left-hand images of the starting simulant and conspicuous by its near-absence in the right-hand images of the ending sublimation sample.

Although it is not clear how the separation occurred, these results suggest that the sample is blue because magnetite has been selectively removed from either the regolith simulant or the nebulizer residue and that sublimation is not necessary. This hypothesis should be tested by magnetically separating magnetite from the simulant and the nebulizer residue and measuring the spectra of the non-magnetic fractions.

(4) If either of these spectra are blue, this would imply that neither sublimation nor Rayleigh scattering have anything to do with Ceres' blue colored areas, which can be made by simply removing a dark absorber from the regolith. We are left with the problem of how such a separation of dark material occurs on Ceres. Unfortunately, a definitive answer is not possible because the identity of the dark material is unknown. The obvious suggestion is separation during settling while the ejecta slurry is still liquid. This is unattractive for two reasons. The first is the weak gravitational field of Ceres. The second is the explosive evaporation, which would not allow time for selective settling.

Another possibility is that the subsurface ice on Ceres was originally emplaced as a liquid, which remained unfrozen long enough for separation by differential settling, or for chemical conversion to a non-dark phase, to occur. This would imply that the blue color is an intrinsic property of a subsurface layer that has been exposed and ejected by the impact.

The Editor may reveal my identity to the authors: Bruce Hapke

Dear reviewers,

We are grateful for your effort and time you spent to carefully read and evaluate our paper. We hope that we were able to address all your concerns. All changes to the manuscript are indicated in the marked version by the red color (new text) and strikethrough (deleted text). Note that we also made changes to the title, which could not be indicated in the marked version. The following are our responses to your questions and comments in a question ("Q") and answer ("A") format.

Reviewer #1 (Roger Clark):

Q: Page 2:

The authors say (lines 47-48) that "Rayleigh scattering inside a particulate surface has not yet been adequately modelled." But Clark et al., 2010 (reference below) did just that, both analogs and radiative transfer modeling. Brown (2014) developed the problem further.

A: Our statement was based on a line in the abstract of Brown (2014): "the necessary mathematical modeling of this phenomenon has not yet achieved maturity". We changed the line to "Models incorporating Rayleigh scattering have since been developed to interpret observations of bluing on other planetary surfaces", now also referring to the Clark+ (2012) paper. Admittedly, we had not fully appreciated the relevance of that paper for our results, but now we make good use of it.

Q: Lines 126-128:

On Tuesday 23 April, we added 0.581 g material to 58.5 mg ultra-pure water in a glass beaker on a magnetic stirrer to achieve a 1% suspension by weight.

0.581 g material to 58.5 mg would be water/material = 10% water

Do you mean 58.5 grams, not mg?

A: We meant 58.5 ml = 58.5 g. Thanks for catching this unfortunate typo.

Q: Page 6 lines 236-239:

"Sublimation introduced more intense bluing for the nontronite than for the analogue material, which suggests that the phyllosilicate foam is predominantly responsible for the blue colour, perhaps through Rayleigh scattering [5, 27]."

Bluing is caused by sub-micron scattering, e.g. sub micron particles in an intimate mixture (Clark et al., 2010). So the experiment only suggests that dispersal of fine particles or sub-micron scattering centers is responsible, not necessarily that sublimation is responsible, Such scattering, for example, could be created by micrometeoroid bombardment, as previously noted in the manuscript.

A: The process that we argue to be directly responsible for the bluing is the sublimation of the ice-phyllosilicate mixture formed on impact, which creates a highly porous network of sub-micron scattering centers whose spacing is sufficiently wide for Rayleigh scattering to occur. We added some clarifying sentences in Sec. 3.3, now referring to Clark+ (2012). We also introduced a comparison with aerogel, which is an example of a porous material that appears blue because of Rayleigh scattering. We believe that this, in combination with the new Fig. 7d, strengthens our case. Note that we modified the title of the paper to emphasize that porosity may be directly responsible for the bluing (and sublimation indirectly).

That said, it is certainly true that one can come up with alternative mechanisms for blue scattering. The

mechanism mentioned by the reviewer, scattering by sub-micron centers created by micrometeoroid bombardment, comes with its own problems. Bombardment by micrometeoroids is a continuous and ongoing process experienced by the entire surface of Ceres. But the blue color is only present around fresh craters and disappears over time. If micrometeoroid bombardment creates such scattering centers, why is all of Ceres not blue? Instead, we argue that the micrometeoroid bombardment gradually destroys the porous structure, leading to fading of the blue color. We now highlight this in the discussion. Last, micrometeorite bombardment as recorded by the Moon can produce significant bluing, but we recognize that the effect depends on the velocity of the impacting particles and the volatile content of the target, limiting the usefulness of this lunar analogy. We invite the reviewer to evaluate our revision.

Q: Page 7, lines 297-298:

"Likewise, micrometeorites might also barely interact with the top layer of the residue."

Why? A micrometeorite will "interact" with the surface on down until all the energy is deposited. To get to the subsurface, the upper surface layers must be penetrated, thus broken up. And breaking up the surface, liberating embedded fine particles would increase the scattering, thus increase bluing.

A: The analogy would be to shoot bullets at a brick and a sponge. The bullet would shatter the brick into pieces. The interaction of the bullet with the sponge would merely be the creation of a bullet-sized tunnel. The bullet would go straight through the sponge and deposit the majority of its kinetic energy in any object behind it. Only few fine particles would be liberated by the impact into a sponge, and they would not be efficiently distributed over the surface, facing difficulties to escape the tunnel. In practice this was demonstrated by the Stardust mission, where tiny particles were found embedded in aerogel at the end of tunnels that were long on the scale of the particles. The porosity of the experimental sublimation residue may have approached that of aerogel (see new microscope image added). Of course, the actual circumstances on Ceres may be different, so we used the word "might". We added a reference to the aerogel penetration tracks in the text. Also, the argument only holds for compact impactors, so we added the word "compact".

Q: Page 7, lines 304+

The experiment does not prove "the leading candidate responsible for the blue color of fresh craters on Ceres is phyllosilicate structure on a sub- μm -scale." It only gives another possible explanation.

I refer the authors to:

Clark, R. N., D. P. Cruikshank, R. Jaumann, R. H. Brown, K. Stephan, C. M. Dalle Ore, K. E. Livo, N. Pearson, J. M. Curchin, T. M. Hoefen, B. J. Buratti, G. Filacchione, K. H. Baines, and P. D. Nicholson, 2012, The Composition of Iapetus: Mapping Results from Cassini VIMS, *Icarus*, 218, 831-860.

There modeling and analog experiments were done to better understand the conditions where particles scatter vs absorb. Bluing and Rayleigh scattering is a diffraction phenomenon. The results in Clark et al show that bluing occurs when a tiny particles with a different index of refraction is separated from the host matrix (an intimate mixture like salt and pepper). If the different index of refraction particle is embedded in another matrix so that the index of refraction difference is smaller, scattering is decreased, e.g. pepper inside a salt grain, then the particle can absorb more than scatter. In space weathering we see this effect: if the iron particles are separated from the matrix (salt and paper mixture), we see bluing (see Figure 17 of Clark et al, and models in Figure 21). Embedded particles show absorption, e.g. see Clark et al, Figures 23a, 23b. Further, the bluing occurs when they scattering centers are widely separated. If the sub-micron particles become too dense, then scattering by diffraction is no different than scattering by larger particles and the surface reflectance begins to look like a slab. For example, 10 micron grains vs 1-micron grains, vs 1 nm grains in contact. Visible and UV photons will not see 1 nm grains any different than a solid slab of the same material.

A: We appreciate the explanations offered by the reviewer. The paper he quotes is indeed highly relevant and we now refer to it. It is certainly true that our experimental results do not prove our hypothesis, but we think we make a good case for it. The challenge we face in the discussion is that alternative explanations for bluing on Ceres are scarce and not clearly formulated in the literature, so we have to do the latter ourselves. The reviewer's comments are helpful in this respect. Our language was perhaps too strong, so we toned it down.

Q: The authors claim that the bluing is caused by "phyllosilicate structure on a sub- μm -scale." However they do not show high enough magnification to evaluate that. The magnification needs to be high enough to show sub 200 nm detail, and better yet, sub 50 nm detail. For example, add to figure 8, images with 1 micron scale bar that is larger than the 50 micron scale bar in the present figure.

A: Done in Fig. 7d. While we understand the rationale for this request, we do note that in Clark+ (2012) it is "sub-0.5- μm diameter particles" that are responsible for Rayleigh scattering (mentioned twice in the abstract), and structures of that size are already visible in Fig. 7c. In that light, asking for detail 10 times smaller ("sub 50 nm") seems a little excessive.

Q: Sub-micron phyllosilicate particles embedded in a salt matrix will have a small index of refraction difference and show absorption more than scattering. It would be sub-micron particles outside the matrix that predominantly contribute to bluing. Further, salt along with sub-micron structure will not contribute significantly to bluing; it will mostly brighten the surface. Complication of this, which has not been previously is the effect of porosity on bluing. This is another area where the authors can contribute significantly to understanding this phenomenon.

A: We agree that any explanation involving salt would be problematic because it would brighten the surface too much. We now briefly touch on this subject in the introduction. Our paper is an example of a study of the relation between porosity and bluing, hence the change in the title. Another example of porosity leading to bluing by Rayleigh scattering is aerogel, to which we now refer in the text.

Reviewer 2 (Kerri Donaldson Hanna):

Q: The manuscript and supplementary materials provide a well-described methodology for the laboratory experiments and the spectral measurements. The nature of these laboratory experiments are quite complex and require bespoke facilities for creating the analogue-ice mixtures, holding the samples at Ceres-like conditions, and collecting spectral measurements over time. Thus, it would be quite difficult for these experiments to be replicated elsewhere. However, I do think the manuscript provides the necessary discussion to understand the experimental set-up and laboratory spectra. While the scope of the manuscript is not to quantify the depths of any of the spectral bands, it would be helpful to include a figure or discussion on the variability of spectral signature over the ~50 minutes needed for each measurement. Adding error bars to one of the spectra, showing that variation would be helpful in demonstrating any changes with time (if any).

A: The reviewer is right to point out the need to better discuss the measurement uncertainties. Each spectrum is recorded with an integration time of 3 seconds per wavelength repeated 10 times (an entire spectrum has 191 wavelengths). We used the average of these 10 measurements as our final reflectance factor, and adopted the standard deviation of those measurements as the uncertainty. For the worst case (the original Ceres analog sample, which was very dark), the uncertainty was smaller than the width of the drawn line over the entire wavelength range (now mentioned in the caption of Fig. 4). That covers

the instrumental uncertainty. With overhead, the acquisition of a single spectrum took take about 50 minutes. With one spectrum recorded per hour, measuring was near-continuous. All this time, the sample would evolve and the spectrometer would record the resulting spectral changes as it progressed along the wavelengths. This means that the depth of the ice bands would typically decrease during the acquisition of the spectrum, leading to an asymmetric band appearance. This variability is probably best assessed by comparing successive spectra and/or by modeling. We do not clearly see how to communicate this in a figure, but we now address this matter in Sec. 3.1. We now also address the issue of stability in the text. Before the start of the sublimation phase under high vacuum, we had recorded a spectrum under medium vacuum. Under those conditions, only minimal sublimation is expected and the reflectance of the icy sample should be stable. We now include this first, stable spectrum in Fig. 6. Furthermore, we now point out that the final spectra were all very similar and can therefore also be regarded as stable.

Reviewer #3 (Bruce Hapke):

Q: (1) The Ceres regolith simulant is chemically unrealistic. Both magnetite and nontronite contain ferric iron, but conditions on Ceres are not expected to be oxidizing, so neither mineral is likely to occur there.

A: Our Ceres regolith simulant is not a chemical/compositional analog but it is a spectroscopic analog. It is a mix of minerals/fluids that give us a spectrum comparable to that which Dawn/VIR detected on the Ceres surface. We measured our Ceres regolith analog with the spare of the VIR instrument in the same range as the flight model (0.25-5 μm). It shows spectral characteristics comparable to Ceres' average spectrum, and is one of many analog mixtures that we produced. Having said that, it is not that we are not confident about the compositional aspect. In fact, VIR on Ceres detected the presence of an Mg-phyllsilicate that was attributed to antigorite (which is present in our mixture), probably produced by the serpentinization process. This is by definition an oxidizing process. As you know, it is a metamorphic process at low pressure and temperature, in which the mafic-ultramafic rocks are transformed through hydrolysis and oxidation processes into serpentinitic rocks. Serpentine minerals are formed starting from olivine through various reactions, some of which are complementary. Olivine is a solid solution between the pure terms forsterite and fayalite that, in the presence of water, start to hydrate, with reactions like these:

fayalite + H_2O \rightarrow magnetite + silica + hydrogen

and

forsterite + H_2O + silica \rightarrow serpentine

These two serpentinization reactions exchange silica between forsterite and fayalite to form serpentine minerals and magnetite. For this reason, even the presence of magnetite, which we use as a darkening agent in our analog, is plausible.

In addition, the presence of nontronite, which is present in our analog mixture as an NH_4 -bearing mineral and key to our findings, is also completely acceptable when we consider that also the formation of this mineral could be linked to a weathering process of ultramafic rocks.

All this to say that even if conditions on Ceres are not expected to be oxidizing, is it impossible to exclude that Ceres did not have different conditions at some point in its history.

Q: (2) The experiment does not realistically simulate conditions on Ceres. In the experiments the slurry remained in liquid form for some time while being disaggregated and nebulized. However, when liquid water is exposed to a vacuum it does not just sit quietly, but evaporates explosively. On Ceres the slurry would boil explosively as soon as it is melted during the impact, rapidly dispersing the solid components in all directions and preventing the formation of the submicron structures. A thick deposit of slurry might remain liquid for a short time while running downhill and could form lobate structures, but it would be boiling violently the whole time.

A: It is very challenging to realistically simulate conditions on Ceres. But a recent experiment reported by Broz+ (2020) simulated a mud flow under Martian conditions. It turns out that, under such low-pressure conditions, a frozen crust quickly forms on the surface of a flowing mud by evaporative cooling, akin to basaltic lavas flowing on Earth, despite vigorous boiling. The authors noted that this process should be even more efficient in the vacuum of Ceres. The experimental surface was found to be frozen, i.e. not desiccated but icy. The porous structures with sub-micron scatters can form by sublimation from a frozen crust. The mud immediately below the crust will freeze at leisure, providing a reservoir for later sublimation after small impacts.

Vigorous boiling was observed yet did not impede formation of the frozen crust. If anything, boiling could help to disperse the mud into small icy particles, perhaps aided by the Mpemba effect. We therefore do not expect boiling to prevent the formation of a porous sublimation residue. We now refer to the Broz+ (2020) experimental results in Sec. 4.

Q: (3) Figure 9 shows the spectra of the original Ceres regolith simulant and the sample remaining after the sublimation process. In the visible the sample has a high reflectance and a steep negative spectral slope, which causes the blue color, while the simulant has a low reflectance with a much smaller negative slope, and is dark. Inspection of the spectra suggests that the simulant spectrum can be made by adding a dark substance with a relatively flat spectral reflectance to the sublimed sample. Magnetite is the obvious candidate for this substance.

Conversely, this inference means that the treatment the simulant received, suspension in water, disaggregation, sedimentation, nebulizing, freezing and subliming, somehow resulted in the selective loss of magnetite relative to the other constituents. (For example, lines 138-143 state that magnetite particles were magnetically separated from something, but it is not clear what that something is. Was it the stirrer magnet, the beaker residue, or what? Why was this done? What was done with this magnetite? Was it mixed back in with the residue, discarded, or what?)

This inference is supported by figure 8; magnetite is conspicuous by its abundance in the left-hand images of the starting simulant and conspicuous by its near-absence in the right-hand images of the ending sublimation sample.

Although it is not clear how the separation occurred, these results suggest that the sample is blue because magnetite has been selectively removed from either the regolith simulant or the nebulizer residue and that sublimation is not necessary. This hypothesis should be tested by magnetically separating magnetite from the simulant and the nebulizer residue and measuring the spectra of the non-magnetic fractions.

A: We stress that the magnetite was not absent from the sublimation residue, but was recovered and added back to the icy sample prior to sublimation. In the text (Sec. 2.2) we had tried to explain the fate of the magnetite, but it seems we could have expressed ourselves more clearly: We used a stirrer with a small magnet to suspend the Ceres analog sample in water. The magnetite immediately attached itself to the stirrer magnet. Other solids also refused to stay suspended. The solids were dried in an oven and momentarily kept apart. The stirrer magnet with the magnetite was also dried. We then separated the magnetite from the stirrer magnet using another, stronger magnet and a piece of paper. This could be

done with minimal loss of magnetite particles, thanks to the attractive properties of magnetite. The magnetite was then added back to the other residual mineral powder. In the mean time we had produced icy particles from the suspension (i.e. nontronite without magnetite and other minerals that did not suspend well). The residue (recovered minerals plus magnetite particles) was added back to the icy particle mixture inside the freezer. We made sure that the final mixture had the same mass fractions of all components as the original Ceres analog sample. We briefly stirred the final sample (everything pre-cooled by liquid nitrogen) to make sure it was well mixed.

Thus, the magnetite was part of the final icy sample. The icy sample was an intra-mixture, meaning ice particles were mixed with solid particles (including magnetite). The interpretation of Fig. 8 is not as straightforward as the reviewer suggests. Figures 8a/c show a flat, thin layer of solid particles. Figures 8b/d, on the other hand, show a porous residue with three-dimensional structure. The elemental map (d) has little depth and shows only the top layer; note how the deeper residue layers at left seen in (b) are not recognized in (d). The volume of the phyllosilicate (color-coded blue) in the residue is very large, as it is now in the form of fluffy spheres instead of compact particles. So compared to (c) there are fewer magnetite particles (color-coded red) apparent in (d). Because the volume of the residue is much larger and the mass fraction of the magnetite is the same, the number density of the latter is much lower. This explains why (d) shows fewer magnetite particles than the reviewer had expected. We now explain this in Sec. 3.2.

One can also distinguish abundant dark particles in the optical microscope image of the sublimation residue in Fig. 7b, most of which (~80%) are magnetite.

Q: (4) If either of these spectra are blue, this would imply that neither sublimation nor Rayleigh scattering have anything to do with Ceres' blue colored areas, which can be made by simply removing a dark absorber from the regolith. We are left with the problem of how such a separation of dark material occurs on Ceres. Unfortunately, a definitive answer is not possible because the identity of the dark material is unknown. The obvious suggestion is separation during settling while the ejecta slurry is still liquid. This is unattractive for two reasons. The first is the weak gravitational field of Ceres. The second is the explosive evaporation, which would not allow time for selective settling.

Another possibility is that the subsurface ice on Ceres was originally emplaced as a liquid, which remained unfrozen long enough for separation by differential settling, or for chemical conversion to a non-dark phase, to occur. This would imply that the blue color is an intrinsic property of a subsurface layer that has been exposed and ejected by the impact.

A: The magnetite was present in our sublimation residue in the same abundance (by weight) as in the original material, but its role was diminished by the large increase of volume through the expansion of nontronite. There was only a separation of magnetite in the sense that magnetite particles could not be brought inside the ice particles, but we mixed them back in with the ice particles. The formation of a porous phyllosilicate "foam" greatly enlarged the total volume of the sublimation residue; we estimate by a factor 5-10 from the thickness of the layer and the volume of the original sample. This volume increase decreased the number density of the magnetite particles. Therefore, the role of magnetite as an absorber of light was reduced in the sublimation residue. This is why the residue was brighter than the original material, and we could more clearly see the intrinsic reflective properties of the phyllosilicate foam. We added this explanation at the start of Sec. 3.3. We revisited our numbers for the concentrations of the constituent minerals, and determined that the ice particles essentially contained only nontronite (now explained in Sec. 2.2). The other components would suspend merely in trace amounts, but were recovered and added back to the ice mixture. That means that the reflective properties of the residue are mostly determined by that of its most voluminous component: the porous

nontronite. We also ran the experiment with pure nontronite, described in the supplementary materials (SM), and found clear bluing for its sublimation residue. So it is consistent that the Ceres analogue residue also displays bluing, because, as said, its reflective properties mostly derive from the expanded nontronite.

Had we know of the reviewer's concerns before the experiment, we would first have selectively removed the magnetite from the analog sample and measured its reflective properties. Unfortunately, the sample is now gone. But the figure below shows the spectra of its individual components as we had measured them prior to the experiment (except nontronite, as its spectrum is shown in the SM). You can verify that none of the components are blue (i.e. have a negative spectral slope over the entire wavelength range shown). It would be miraculous if put together in a dry mixture minus magnetite, a blue color would suddenly emerge. Finally, we stress that bluing after sublimation was already observed for the phyllosilicate montmorillonite by Poch+ (2016). In that case sublimation also resulted in a porous network. Our results are fully consistent with theirs.

The spectra in this figure will be included in an upcoming paper by the co-authors who prepared the Ceres analogue sample. For that reason the plot is not included in our paper, and we show these spectra for the exclusive use in the current review process.

REVIEWER COMMENTS

Reviewer #2 (Remarks to the Author):

In this manuscript, the authors describe in detail laboratory sublimation experiments of a Ceres-like analogue (based on the Dawn spectral observations). These experiments include the spectral characterisation of the analogue sample as it sublimates to characterise the visible to near infrared reflectance changes over time and the characterisation of the final residue. The manuscript clearly describes the experiment and the resulting spectra (including the quantification of uncertainties in the spectra) and their implication for the bluing of Ceres' surface.

These experiments are quite novel and can only be done by a very small subset of laboratory facilities with the right environmental chambers. While it would be difficult for others to easily replicate these measurements, the manuscript clearly describes the techniques used.

I believe this manuscript to be of great interest to the community.

Kerri Donaldson Hanna

Reviewer #3 (Remarks to the Author):

Comments on "Bluing on Ceres linked to porosity resulting from sublimation" (revised) by Shroder et al:

Reviewer: Bruce Hapke

The authors have satisfactorily addressed the concerns I expressed in my review of the original manuscript. I have one minor change that I would like to see and then the paper will be acceptable for publication. The authors attribute the increased albedo of the sublimation residue over the starting Ceres analog to increased volume. However, increasing the porosity of a material decreases the reflectance, not increases it (Hapke 2016, *Icarus*, 273, 75-83). The increased albedo is due to the decreased size of the phyllosilicate particles.

To understand this, consider the material to be composed of a mixture of two materials: the phyllosilicates, denoted by subscript p , and the non-phyllosilicate remainder, denoted by n . The

reflectance of a material is controlled by its single scattering albedo, W . For a mixture of materials W is given by (Hapke, 2012, Theory of Reflectance and Emittance Spectroscopy, 2nd ed., Cambridge U. Press):

$$W = (F_p * W_p * Q_{ep} / D_p + F_n * W_n * Q_{en} / D_n) / (F_p * Q_{ep} / D_p + F_n * Q_{en} / D_n) ,$$

where F_p is the filling factor of the phyllosilicates (the fraction of the volume occupied by phyllosilicates) and F_n is the filling factor of the remainder material, W_p and W_n are their single scattering albedos, Q_{ep} and Q_{en} are their extinction efficiencies, and D_p and D_n are the mean particle sizes of the phyllosilicate and the remainder, respectively. Judging from figure 8, in the starting material $D_p \sim 50$ micrometers and $D_n \sim 10$ micrometers. W_p was probably around 50%, while W_n was smaller because of the magnetite. The albedo was low, which implies that the n terms dominated W .

Assume that all the materials remained well mixed and no separation occurred during the sublimation. Then, after the sublimation was complete, F_n and F_p both decreased by the same factor of 5 – 10, but this did not affect the ratio W . Also, W_n and D_n did not change. What did change was D_p . The particles in the sublimate were filaments about 100 nm in diameter and 1000 nm or so in length, so D_p decreased by two orders of magnitude. The decrease in size caused W_p to increase. Because of the major decrease in phyllosilicate particle size in the sublimate, the p terms are much larger than the n terms, and the p terms dominate W , increasing the reflectance.

The authors assume that the bluing is caused by Rayleigh scattering. The individual filaments are certainly small enough to do this especially in the near IR, but the filaments must also be sufficiently separated that they can act as independent scatterers, otherwise the light waves see closely-packed scatterers as one single object larger than the wavelength. It is not clear from the images that this is the case. However, the authors claim that pure sublimated nontronite is blue, which supports their assumption that the filaments do scatter independently. It would be nice if the authors would add a figure showing the before and after spectra of nontronite.

Bruce Hapke

Additional consultation comments of Reviewer #3 (Remarks to the Author):

"As you requested, I have read the comments by reviewers #1 and #2. While I cannot speak for them, in my opinion, the authors have satisfactorily addressed the reviewers' comments, and I recommend that the paper be accepted for publication."

Dear reviewers,

We are grateful for your effort and time you spent to carefully read and evaluate our paper. We especially thank Dr. Hapke, who was so kind to also evaluate our response to reviewer #1. This time, only Dr. Hapke had comments, to which we reply below. All changes to the manuscript are indicated in the marked version by the red color (new text) and strikethrough (deleted text). Note that we made small changes to the title and abstract. The following is our response to Dr. Hapke's comments in a question ("Q") and answer ("A") format.

Reviewer #3 (Bruce Hapke):

Q: The authors attribute the increased albedo of the sublimation residue over the starting Ceres analog to increased volume. However, increasing the porosity of a material decreases the reflectance, not increases it (Hapke 2016, *Icarus*, 273, 75-83). The increased albedo is due to the decreased size of the phyllosilicate particles.

To understand this, consider the material to be composed of a mixture of two materials: the phyllosilicates, denoted by subscript p, and the non-phyllosilicate remainder, denoted by n. The reflectance of a material is controlled by its single scattering albedo, W. For a mixture of materials W is given by (Hapke, 2012, *Theory of Reflectance and Emission Spectroscopy*, 2nd ed., Cambridge U. Press):

$$W = (F_p * W_p * Q_{ep}/D_p + F_n * W_p * Q_{en}/D_n) / (F_p * Q_{ep}/D_p + F_n * Q_{en}/D_n) ,$$

where F_p is the filling factor of the phyllosilicates (the fraction of the volume occupied by phyllosilicates) and F_n is the filling factor of the remainder material, W_p and W_n are their single scattering albedos, Q_{ep} and Q_{en} are their extinction efficiencies, and D_p and D_n are the mean particle sizes of the phyllosilicate and the remainder, respectively. Judging from figure 8, in the starting material $D_p \sim 50$ microns and $D_n \sim 10$ microns. W_p was probably around 50%, while W_n was smaller because of the magnetite. The albedo was low, which implies that the n terms dominated W.

Assume that all the materials remained well mixed and no separation occurred during the sublimation. Then, after the sublimation was complete, F_n and F_p both decreased by the same factor of 5 – 10, but this did not affect the ratio W. Also, W_n and D_n did not change. What did change was D_p . The particles in the sublimate were filaments about 100 nm in diameter and 1000 nm or so in length, so D_p decreased by two orders of magnitude. The decrease in size caused W_p to increase. Because of the major decrease in phyllosilicate particle size in the sublimate, the p terms are much larger than the n terms, and the p terms dominate W, increasing the reflectance.

A: We respectfully disagree with Dr. Hapke for the reason that separation of materials did, in fact, occur. The equation is valid for an intimate mixture of "p" (phyllosilicate) and "n" (mostly magnetite) particles. After sublimation, the phyllosilicate scattering centers (filaments) were very small, as can be seen in our Fig. 7, but the "p" particles are not these filaments. During the experiment we mixed the magnetite particles with the icy spheres. Sublimation of the ice led to the formation of porous phyllosilicate spheres of 50 μ m size. Magnetite particles were therefore not present within the spheres, and it is these spherical "particles" that were mixed with the magnetite particles. Therefore the fundamental "p" particles in the equation are the porous phyllosilicate spheres, and not the tiny filaments. The mixing equation for our sample is:

$$w = (A_p w_p + A_n w_n) / (A_p + A_n), \text{ with } A_x = N_x \sigma_x Q_{E,x},$$

with number density N and cross-section σ (Hapke, 2012). We assume that w_p did not change much

compared to the original phyllosilicate particles and that w_n was unchanged. Then we must compare the $A_p = N_p \sigma_p Q_{E,p}$ and $A_n = N_n \sigma_n Q_{E,n}$ terms before and after sublimation. As Dr. Hapke correctly noted, the original (solid) phyllosilicate particles were about the same size as the porous phyllosilicate spheres ($\sim 50 \mu\text{m}$). But after sublimation, there must have been many more phyllosilicate particles (low density spheres), as the sample had increased in volume by a factor 5-10. That means that after sublimation, σ_n is the same, σ_p is about the same, but N_p and N_n are substantially higher and lower, respectively. The porous spheres have abundant internal scattering centers and therefore $Q_{E,p}$ has increased compared to the original, solid phyllosilicate particles through increase of $Q_{S,p}$ in $Q_{E,p} = Q_{A,p} + Q_{S,p}$. On the other hand, $Q_{E,n}$ is still the same. All this means that A_p had increased after sublimation compared to A_n , and **w is thereby larger**, mostly because of the **decrease in N_n** and **increase in $Q_{E,p}$** . Undoubtedly, some of the details are not entirely accurate, but we believe that these are the dominant factors affecting light scattering by the samples.

Previously, the text only mentioned the increase in volume of the sample as an explanation for the albedo increase. Now we also mention the increased scattering efficiency of the phyllosilicate spheres in Sec. 3.3 and added a cartoon of light scattering by the samples as Fig. S6 in the supplementary material.

Q: The authors assume that the bluing is caused by Rayleigh scattering. The individual filaments are certainly small enough to do this especially in the near IR, but the filaments must also be sufficiently separated that they can act as independent scatterers, otherwise the light waves see closely-packed scatterers as one single object larger than the wavelength. It is not clear from the images that this is the case. However, the authors claim that pure sublimated nontronite is blue, which supports their assumption that the filaments do scatter independently. It would be nice if the authors would add a figure showing the before and after spectra of nontronite.

A: We fully agree and that is why we had already included such a plot in the Supplementary Material (Fig. S3). Please note that the nontronite experiment primarily served as a practice run for the Ceres analog experiment, as this experiment was technically very challenging. The results in terms of the spectral changes for pure nontronite are more difficult to interpret because of the high reflectance of the sample and thereby abundant multiple scattering. Nevertheless, we found that the post-sublimation nontronite sample was bluer than the original sample. The details of the nontronite experiment are described in the Supplementary Material. We now refer in the main text to Fig. S3.

REVIEWER COMMENTS

Reviewer #3 (Remarks to the Author):

Comments on “Bluing on Ceres’ surface due to high porosity resulting from sublimation” (revised) by S. Shroeder et al

Reviewer: Bruce Hapke

The paper and the response to my original comments asserts that the high albedo and bluish color of spots on Ceres is due to high porosity. The arguments presented in support of this are incorrect. They would be valid if the authors were examining the surface with a microscope that allowed them to see only a small area on the surface. However, that is certainly not the case when Dawn viewed Ceres.

Assume that a detector views an area on the surface of a medium. Assume that the detector’s field of view is large enough to be representative of the whole medium. The field of view includes a large number of both light and dark particles, resulting in a certain reflectance that is measured by the detector. Next, expand the medium in all directions by increasing the distance between particles. This reduces the number of dark particles viewed by the detector, and the authors argue that this increases the reflectance. However, the expansion also reduces the number of light particles viewed, so the ratio of light-to-dark particles remains the same and the reflectance is unchanged.

(Actually, this is not quite true. Increasing the porosity allows the incident light to penetrate deeper into the medium, where it has a lower probability of escaping, so the reflectance is decreased. This is well understood theoretically and has been verified experimentally.)

The reason the reflectance is increased by the sublimation is that each relatively large phyllosilicate particle in the starting mixture has been converted into a large number of relatively smaller particles with higher single scattering albedos, thus increasing the light-to-dark ratio. The fact that the medium expanded is irrelevant. The title and discussion in the paper need to be revised.

The next question is “What causes the bluing?”. The authors claim that it is caused by the spheres. However, although it is somewhat difficult to tell from the images in figure 7, it seems to me that the

structural elements of the spheres are close enough together that they would not scatter independently. The light would view a sphere as a single particle and Rayleigh scattering would be negligible. The text describes the spheres only as “pale” with no mention of a bluish color, whereas a blue color should be obvious if the spheres were the only source of blue in the residue.

The only components of the residue that appear to be small enough and sufficiently separated that they could act like Rayleigh scatterers are the filaments in the lacey matrix. What is the origin of this material? From figure 8 it appears to be phyllosilicate and distributed throughout the matrix. Or is it associated only with the spheres? Is it some kind of coating that formed on the spheres and then peeled off? Answers to these questions are important if the lacey matrix is the source of the color.

I apologize for not being clear when I said that I would like to see a figure showing the results of the nontronite-only experiment. What I meant was that I believe a figure showing the before and after spectra was important enough that it should be included in the paper and not relegated to separate supporting material.

Bruce Hapke

Dear Dr. Hapke,

Thank you for providing us with feedback to our paper on the bluing experiment once more. We realized that we had not fully understood your previous review. We agree with you that porosity, as you described it, plays no role in the albedo increase of the sublimation residue. But this definition of sample porosity does not correspond to our experiment. So, there may have been a misunderstanding about the role and nature of porosity in our samples. Let us try to explain better (your words in italics):

Assume that a detector views an area on the surface of a medium. Assume that the detector's field of view is large enough to be representative of the whole medium. The field of view includes a large number of both light and dark particles, resulting in a certain reflectance that is measured by the detector. Next, expand the medium in all directions by increasing the distance between particles. This reduces the number of dark particles viewed by the detector, and the authors argue that this increases the reflectance. However, the expansion also reduces the number of light particles viewed, so the ratio of light-to-dark particles remains the same and the reflectance is unchanged.

That is indeed the usual meaning of porosity of a particulate sample. In our experiment, however, the original solid "light" (phyllosilicate) particles were replaced by an expansive porous network made of nano-sized phyllosilicate filaments and ribbons. The original "dark" particles remained unchanged and were scattered throughout the porous network. The situation was not that the distance between the particles had increased, it was the "light" particles themselves that had rearranged and expanded into a foamy structure. This is not porosity in the sense of your 2008 paper. This complete rearrangement of the phyllosilicates is due to the unique properties of these minerals (more about that below).

The reason the reflectance is increased by the sublimation is that each relatively large phyllosilicate particle in the starting mixture has been converted into a large number of relatively smaller particles with higher single scattering albedos, thus increasing the light-to-dark ratio. The fact that the medium expanded is irrelevant.

That is essentially true. And the small particles (filaments in the phyllosilicate foam) are so small that we expect them to scatter in Rayleigh regime. But for the sublimation residue to show Rayleigh scattering, the filaments need to be sufficiently widely separated. The high porosity of the phyllosilicate foam takes care of that. Thus, the porosity of the sublimation residue plays a crucial role in promoting Rayleigh scattering (more about that below), which is the core message of our paper. The experiment converted the large phyllosilicate particles in the starting mixture into a large number of small particles arranged in a foam. The new particles are small enough to scatter in the Rayleigh regime and widely separated, which leads to both bluing and brightening in the visible.

The only components of the residue that appear to be small enough and sufficiently separated that they could act like Rayleigh scatterers are the filaments in the lacy matrix. What is the origin of this material? From figure 8 it appears to be phyllosilicate and distributed throughout the matrix. Or is it associated only with the spheres? Is it some kind of coating that formed on the spheres and then peeled off? Answers to these questions are important if the lacy matrix is the source of the color.

That is our hypothesis. The origin of the lacy filaments lies in the suspension of the phyllosilicate particles in liquid water. When smectite phyllosilicates like montmorillonite are mixed with liquid water, hydration of the inter-layer space will lead to partial delamination, resulting in suspended nano-sized platelets and aggregates thereof. Freezing the suspension leads to the growth of water ice crystals, with platelets concentrating at the crystal interfaces. Subsequent sublimation replaces the ice crystals with

voids, and only the phyllosilicate platelets remain to form a porous network (foam). This figure explains the process for the Ceres analog material:

The diagram shows the sublimation process for the Ceres analogue material. Inside each spherical ice particle produced through the experimental SPIPA-B protocol, phyllosilicate platelets formed veins between the water ice crystals. During the sublimation of the ice, the water crystals were replaced by voids and the platelets remained to form a porous network. In some cases, the desiccated structure retained the spherical shape of the original ice particle. More often, spheres fragmented into a disordered fluffy medium of platelets (filaments), as seen in the scanning electron microscope (SEM) images below.

We now show a number of SEM images to further illustrate the formation process of the lacey, porous structure. Please note the scale bar at the bottom left of each image. First, we show four SEM images of the pure nontronite sublimation residue. Note how high porosity is pervasive in the foamy superstructure, with spherical shapes still recognizable in many places:

In the Ceres analog sublimation residue, most of the spheres had fused into a porous mass during the sublimation process. This disordered mass reveals what the inside of the spheres looks like. In a few places you can still see the original spheres:

The entire mass, not only the remaining spheres, is spongy and porous. The other mineral particles that would not suspend (mostly magnetite) were mixed with the icy spheres. As such, these particles are not found inside the spheres, but were nevertheless incorporated in the spongy mass:

The next question is "What causes the bluing?". The authors claim that it is caused by the spheres. However, although it is somewhat difficult to tell from the images in figure 7, it seems to me that the structural elements of the spheres are close enough together that they would not scatter independently. The light would view a sphere as a single particle and Rayleigh scattering would be negligible. The text describes the spheres only as "pale" with no mention of a bluish color, whereas a blue color should be obvious if the spheres were the only source of blue in the residue.

We had explained bluing in the Ceres analog residue in terms of the spheres because we thought it was a useful mental picture. In reality, most spheres had broken apart and fused into a porous, foamy mass, as seen in the SEM images above. It is this entire foamy structure that causes the bluing, rather than only the spheres. Where you can recognize the spheres in an optical microscope image (Fig. 7b), they look pale because they are cleaner than the porous mass. The absorbing mineral particles like magnetite could not enter the spheres (they were mixed with the ice particles), but the porous mass integrates them. The porous mass may be less "clean", but it should not be less porous than the spheres because it derives from them. Thus, the high porosity is pervasive, and not constrained to the spheres, which is consistent with the large volume increase that we observed from original powder to sublimation residue. It is the filaments inside both the mass and the remaining spheres that we suggest to scatter in Rayleigh fashion. The porosity of the pure nontronite sublimation residue was extreme, and it showed strong bluing. The porosity of the Ceres analog sublimation residue is also very high, as you can see in the SEM micrographs above. The bluing displayed by this residue is less strong than that of the pure nontronite, but still very obvious (Fig. 9, note different scales for the reflectance ratio). The reason for this is not entirely clear, but must be related to the presence of other mineral particles.

I apologize for not being clear when I said that I would like to see a figure showing the results of the nontronite-only experiment. What I meant was that I believe a figure showing the before and after spectra was important enough that it should be included in the paper and not relegated to separate supporting material.

Thank you for the clarification. We moved a figure from the supplementary material to Fig. 9 of the main paper.

The title and discussion in the paper need to be revised.

In the light of your comments we revised the title to "Bluing on Ceres' surface caused by a foam-like sublimation residue", to avoid any possible misunderstanding about the word porosity. We also replaced Fig. 7c. Furthermore, we added SEM images, including the ones above, to the Supplementary Material. We tried to explain things better throughout the paper (especially in Secs. 3.2 and 3.3) in the hope that we could lay your concerns to rest.

After submitting this report, the editor reminded us that we had not responded to this point in your review report:

The paper and the response to my original comments asserts that the high albedo and bluish color of spots on Ceres is due to high porosity. The arguments presented in support of this are incorrect. They would be valid if the authors were examining the surface with a microscope that allowed them to see only a small area on the surface. However, that is certainly not the case when Dawn viewed Ceres.

We are not sure whether this is still your opinion, after reading our explanation above that aimed to

resolve the misunderstanding about the role of porosity in our experiment. We think that interpreting remote spacecraft observations by means of laboratory experiments is commonplace and not controversial. In the case of our experiment, the laboratory measurement was on the scale of a centimeter. The light scattering processes and structure of the sample are on the order of (tens of) micrometers. A difference of 3 to 4 orders of magnitude. The blue color on Ceres was observed by the Dawn framing camera on a decameter scale (Schröder+ 2017), adding 3 to 4 more orders of magnitude. The Dawn VIR resolution was lower than that of the cameras, but fully consistent when it comes to the blue color (Ciarniello+ 2017, Rousseau+ 2020). If the surface is blue on a scale of centimeters, it is very likely also blue on a scale of decameters. We apologize if we did not fully understand what you meant.

Ciarniello+ (2017) <https://doi.org/10.1051/0004-6361/201629490>

Rousseau+ (2020) <https://doi.org/10.1051/0004-6361/202038512>

Schröder+ (2017) <http://dx.doi.org/10.1016/j.icarus.2017.01.026>

REVIEWERS' COMMENTS

Reviewer #3 (Remarks to the Author):

The authors have satisfactorily addressed the comments in my reviews and the revised paper is recommended for publication.

I have only one minor comment: many paragraphs are too long. The readability of the paper would be improved if the authors would break them up into shorter paragraphs.

Bruce Hapke